# The microbiome of cryospheric ecosystems

Massimo Bourquin [1,5 ✉], Susheel Bhanu Busi [2,5], Stilianos Fodelianakis[1], Hannes Peter[1], Alex Washburne[3], Tyler J. Kohler[1], Leïla Ezzat[1], Grégoire Michoud [1], Paul Wilmes [2,4] & Tom J. Battin [1 ✉]

The melting of the cryosphere is among the most conspicuous consequences of climate change, with impacts on microbial life and related biogeochemistry. However, we are missing a systematic understanding of microbiome structure and function across cryospheric ecosystems. Here, we present a global inventory of the microbiome from snow, ice, permafrost soils, and both coastal and freshwater ecosystems under glacier influence. Combining phylogenetic and taxonomic approaches, we find that these cryospheric ecosystems, despite their particularities, share a microbiome with representatives across the bacterial tree of life and apparent signatures of early and constrained radiation. In addition, we use metagenomic analyses to define the genetic repertoire of cryospheric bacteria. Our work provides a reference resource for future studies on climate change microbiology.

[1] River Ecosystems Laboratory, Centre for Alpine and Polar Environmental Research (ALPOLE), École Polytechnique Fédérale de Lausanne, EPFL, Lausanne, Switzerland. [2] Luxembourg Centre for Systems Biomedicine, University of Luxembourg, Campus Belval, 7, avenue des Hauts-Fourneaux, L-4362 Esch-sur-Alzette, Luxembourg. [3] Selva Analytics LLC, Bozeman, MT 59718, USA. [4] Department of Life Sciences and Medicine, Faculty of Science, Technology and Medicine, University of Luxembourg, 7, avenue des Hauts-Fourneaux, L-4362 Esch-sur-Alzette, Luxembourg. [5]These authors contributed equally: Massimo Bourquin, Susheel Bhanu Busi. ✉email: massimo.bourquin@epfl.ch; tom.battin@epfl.ch

Microorganisms dominate the biosphere, maintain eco- system processes, and play key roles in global biogeo- chemical cycles. The microbiome of cryospheric ecosystems, the nearly 20% of Earth's surface where water remains frozen for at least one month of the year[1], currently figures among the least understood microbiomes on Earth[2–6]. This is noteworthy given that the cryosphere is melting at an unprecedented pace owing to climate change. Motivated by the exploration of life in a planetary context[7] and the discovery of new biomolecules for biotechnology[8], classical microbiology and (more recently) advances in sequencing technologies have unra- velled physiological and molecular processes underpinning the adaptation of cold-adapted bacteria (i.e., psychrophiles) to the cryospheric environment[9,10]. More specifically, metagenomics has provided new insights into the structure and function of complex microbial communities of some cryospheric ecosystems, such as permafrost soils[2,11], leading to a better understanding of the role of these ecosystems in global biogeochemical cycles and their vulnerability to climate change.

However, we are still missing an integrative understanding of the microbiome across the various and often underexplored cryospheric ecosystems on Earth[3,5,6]. Here we present a global catalogue of microorganisms from various cryospheric ecosystems and at a taxonomic resolution that allows detection of cryosphere-adapted lineages and associated traits. We leverage sequence data from numerous published studies ranging from snow to permafrost ecosystems to shed light on the global cryospheric microbiome. While also illuminating geographical biases and underexplored habitats in the currently available cryospheric data, our study con- stitutes an important resource for the study of cryospheric life in general and its potential future in a warmer world.

## Results and discussion

**The dataset**. We curated and explored 695 published 16S rRNA gene samples from cryospheric ecosystems (Methods section and Supplementary Table 7), including polar ice sheets, mountain gla- ciers and their proglacial lakes, permafrost soils and the coastal ocean under the influence of glacier runoff, and compared these to 3552 published 16S rRNA gene samples from non-cryospheric ecosystems, including temperate and tropical lakes and soils (Sup- plementary Table 7). This approach allowed us to identify and explore features specific to the cryospheric microbiome and com- pare it to other environmental microbiomes. However, we note a geographical bias towards polar regions in current publicly available repositories, and the paucity of alpine samples specifically highlights the need to further characterise these habitats given that they are among the most endangered cryospheric ecosystems globally. This bias is further compounded by the inconsistent methodologies applied across studies (e.g. primer pairs and sequencers used). To account for potential primer biases, we analysed two 16S rRNA primer pairs (Primer Pair 1, PP1: 341f-785r; Primer Pair 2, PP2: 515f-806r)[12,13] commonly used in amplicon high-throughput sequencing. In total, this dataset contains 241,502,708 paired sequence reads, resulting in 530,254 and 410,931 amplicon sequence variants (ASVs) for PP1 and PP2, respectively. Moreover, all taxo- nomic analyses were performed at the genus level, to account for the limitations of 16s rRNA amplicon data. To gain deeper insights into the functional space of the cryospheric microbiome, we compared 34 published metagenomes from cryospheric ecosystems with 56 metagenomes from similar but non-cryospheric ecosystems (Fig. 1A). Given the difficulty of obtaining high-quality metagen- omes from cryospheric ecosystems, we restricted our analyses to glacier surfaces, ice-covered lakes, and Antarctic soils. Although our analyses were limited to samples where raw sequence data are available (Methods section), the breadth of habitats covered are

representative of the most abundant cryospheric ecosystems, e.g., glacier ice, cryoconites, subglacial lakes and sea ice. On the other hand, several niches such as glacier snow, glacier-fed rivers/streams, and the full-breadth of permafrost may not entirely be represented due to data unavailability. We reanalysed all metagenomes using the same bioinformatic pipeline (IMP3; see Methods section) to avoid analytical biases. Overall, the metagenomic analyses from 2,427,818,072 paired reads yielded 41,068,842 gene sequences. Thus, we here present a catalogue representing a snapshot of the func- tional diversity in the cryospheric microbiome, integrating across diverse habitats. This represents what we believe to be the first global overview of the functional repertoire of the Earth's cryosphere compared to other ecosystems.

**A cryospheric microbiome**. Given the communal constraints imposed by the harsh environment of cryospheric ecosystems (e.g., low temperature, oligotrophy), we expected them to harbour a specific microbiome. Accordingly, machine-learning classifica- tion (logistic regression models, Methods) based on community composition was able to differentiate between cryospheric and non-cryospheric microbiomes with high accuracy (balanced accuracy >0.96, Supplementary Table 1). Both primer pairs consistently yielded a high classification accuracy and especially a high precision. Interestingly, many of the discriminating cryo- spheric ASVs were spread widely across the bacterial tree of life (Fig. 1A and Supplementary Fig. 1).

The notion that a part of the microbiome is specific to the cryosphere is also strongly supported by phylogenetic analyses of the 16S rRNA gene amplicon dataset. First, we found higher pairwise phylogenetic overlap among cryospheric samples than among cryospheric/non-cryospheric or non-cryospheric samples (Sorensen's index, Fig. 1C; Wilcoxon test, Holm adj. $p < 0.001$). This points towards a phylogenetic diversity that is specific to the cryosphere. Second, when we examined cross-sample nearest taxon distances (β-NTDs), we found that taxa in cryospheric samples have lower β-NTDs in other cryospheric samples than in non-cryospheric samples (Fig. 1D; Wilcoxon test, Holm adj. $p < 0.001$). This was less evident for non-cryospheric samples (Supplementary Table 2). Because phylogeny and functional similarity usually correlate at short phylogenetic distances[14], this finding suggests higher niche similarity for cryospheric bacteria compared to their non-cryospheric equivalents. This evokes specific selective constraints of cryospheric environments acting on taxa across the entire bacterial tree of life. Interestingly, when we further examined radiation patterns, we found that taxa in a given cryospheric microbial community had on average larger phylogenetic distances (α-MPD) than their counterparts in a non- cryospheric community (linear model, $p < 0.001$). This could suggest early radiation events with subsequent "pruning" of phylogenetic diversity, which would explain the observed patterns[15]. However, we cannot exclude nor disentangle the action of contemporary evolutionary and assembly processes that can jointly shape community phylogenies. For example, trans- duction and genome plasticity have repeatedly been linked with cold adaptation in cryospheric bacteria. Moreover, horizontal gene transfer has also been shown to promote the diffusion of cold-adaptation genes[16]. Nevertheless, given the large number and breadth of included cryospheric ecosystems, we posit that the topologies of the inferred phylogenies are less prone to assembly processes. We rather interpret that the observed patterns are signs of early and constrained radiation in the cryospheric microbiome. Collectively, these results point to similar evolutionary trajectories in cryospheric microbiomes, probably owing to similar environ- mental conditions across various cryospheric ecosystems, over timescales, relevant for bacterial macroevolution.

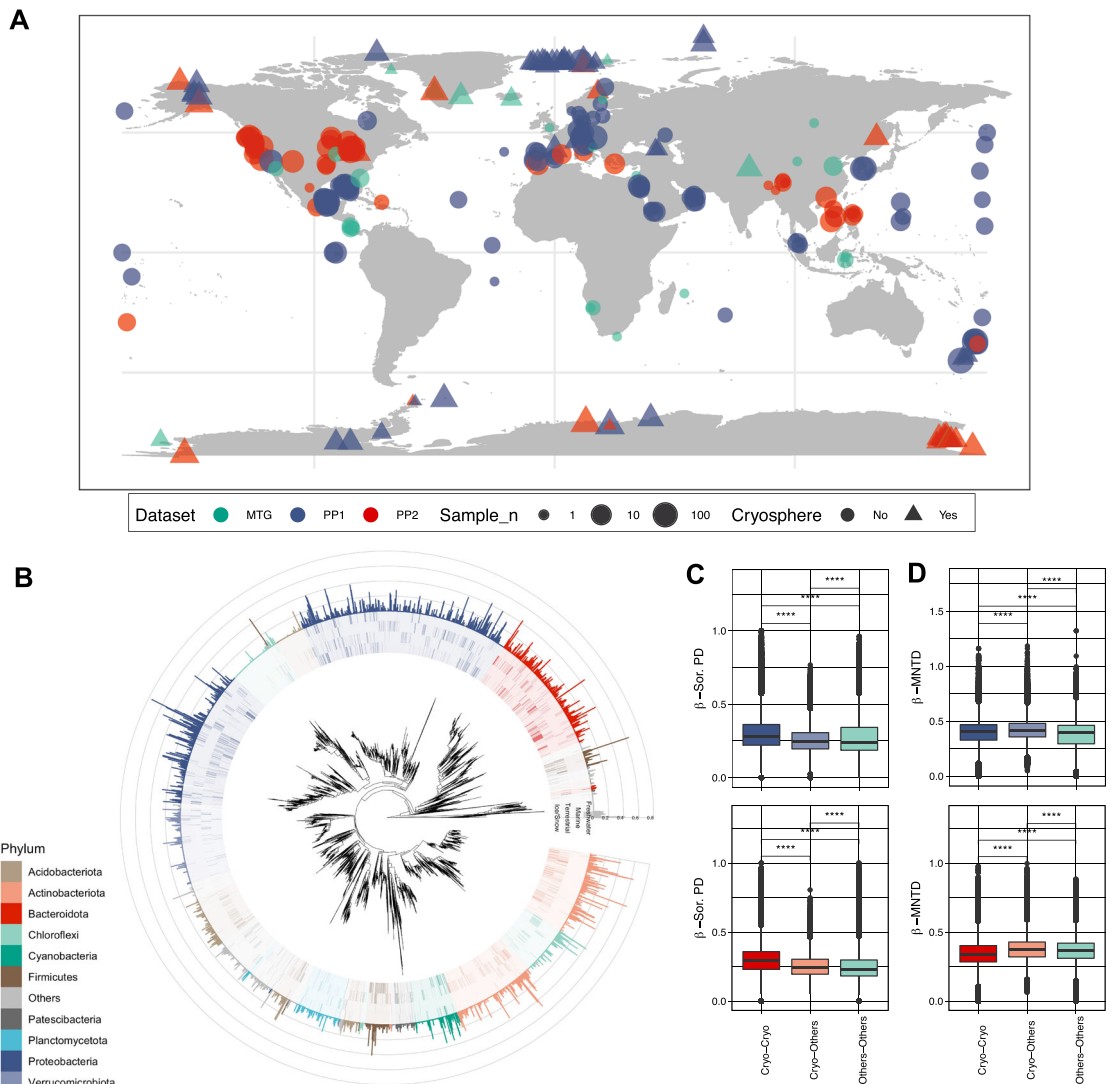

**Fig. 1 A unique cryospheric microbiome. A** Geographic distribution of the 16 S rRNA gene samples for the two primer pairs (PP) and metagenomes for both cryospheric and non-cryospheric ecosystems, where GPS coordinates were available on NCBI. Symbol size denotes the number of samples per site (see Supplementary Table 7). **B** Phylogenetic tree based on abundant ASVs (>0.5% relative abundance in at least one sample) in the PP1 dataset. The heatmap (inner rings) shows the presence (at $a > 0.5$% relative abundance threshold) of ASVs in the four ecosystem types of the cryosphere (ice and snow, terrestrial, coastal ocean and freshwater). The barplot (outer ring) represents the coefficient for the SVM classifier analysis, highlighting discriminating ASVs. **C** Sorensen's phylogenetic index of $\beta$-diversity ($n_1 = n_2 = 84,461$ for PP1, and $n_1 = n_2 = 99,000$ for PP2) and **D** $\beta$-MNTD calculated across pairs of samples in the cryospheric samples (Cryo-Cryo), pairs of cryospheric and non-cryospheric samples (Cryo-Others) and pairs of non-cryospheric (Others-Others) samples (sample sizes are listed in Supplementary Table 2). The top panel (shades of blue) is for PP1, the bottom one (shades of red) for PP2; two-sided Wilcoxon tests were performed to assess significance in panels **C** and **D**; the Holm method was used to correct for multiple testing (****: 0–0.0001). Boxplots depict the median and the 25th and 75th quartiles, whiskers extend to values within 1.5 times the interquartile range, and the remaining points are outliers. Effect sizes and exact p-values are available in Supplementary Table 2. Source data are provided as a Source Data file.

The abundance of a given species in an ecosystem generally reflects its fitness and adaptive capacity to the respective environmental conditions. Therefore, we explored patterns of differential abundance (Methods section) and found 589 bacterial genera with higher abundances in cryospheric compared to non-cryospheric samples (Ancom, W statistic > 0.7, CLR mean difference > 0) that hereafter will be referred to as cryospheric genera. These genera were distributed widely across the bacterial tree of life and encompassed 46 different phyla. Despite this wide distribution, we found that 34.8% and 13.4% of the cryospheric genera were affiliated *Proteobacteria* and *Bacteroidota*, respectively (Fig. 2A). The relevance of *Proteobacteria* is in line with the high prevalence of *Alpha-* and *Gammaproteobacteria* typically reported in the cryospheric

literature[4,17]. Genera belonging to the *Alpha-* and *Gammaproteobacteria* classes displayed the highest differential abundance and included *Sphingomonas, Polaromonas, Rhodoferax, Brevundimonas, and Acidiphilum* (Fig. 2B) — some of them with taxa typically reported to be psychrophiles[10,18–20]. *Bacteroidota* was the second most important phylum of the cryospheric microbiome with *Hymenobacter, Ferruginibacter,* and *Polaribacter* (for instance) as dominant genera, all of which are known from permafrost soils and ice ecosystems[21,22]. Furthermore, as previously reported[23,24], the cryospheric genera included members of the *Actinobacteria, Chloroflexi* and *Cyanobacteria* phyla, alongside some *Firmicutes*. The former two are particularly common in supraglacial environments[4], and *Cyanobacteria* are important components of

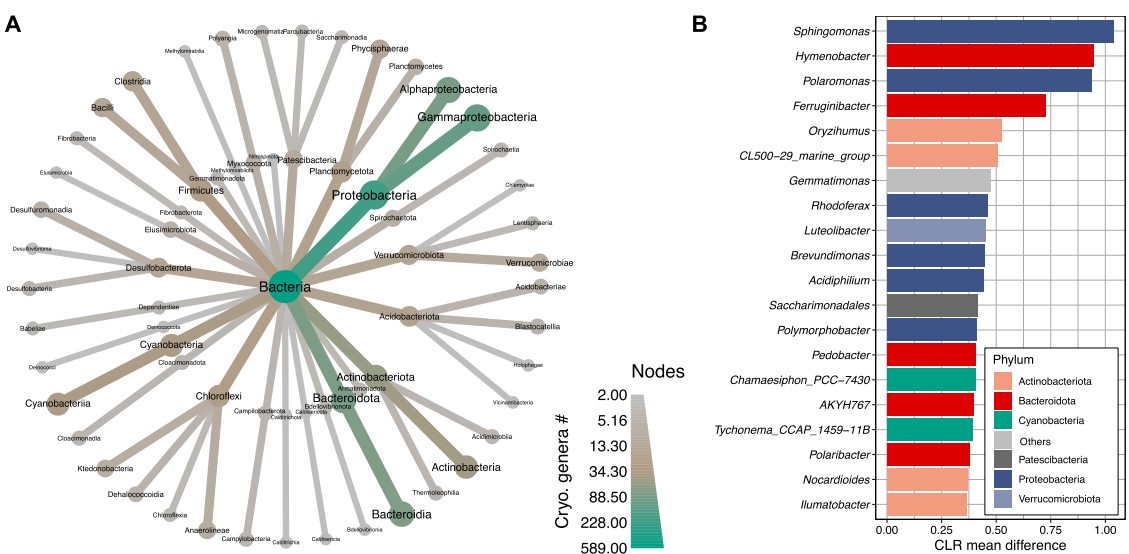

**Fig. 2 Cryospheric genera and shared genomic properties. A** Taxonomic tree representing the number of cryospheric genera per taxon with colours. Only taxa containing at least two cryospheric genera are shown (down to the class level). **B** Bar plot showing the bacterial genera significantly enriched in the cryosphere with the highest centered log-ratio (CLR) mean difference (based on ANCOM analysis). The colours represent the phylum level taxonomic classification. Source data are provided as a Source Data file.

cryoconite microbiomes[25]. Our global analyses thus corroborate and extend previous reports on microbiome composition in distinct cryospheric ecosystems. Furthermore, our differential abundance analysis unveiled genera (e.g., *Oryzihumus* or *Pseudolabrys*) that have not been previously associated with the cryosphere (Fig. 2B). More importantly, many of the detected cryospheric genera only have placeholder names due to the lack of cultivated representatives (e.g., CL_500-29_marine_group, hgcl_clade, TRA3-20), underlining unique bacterial groups that are yet to be described. Collectively, these findings unveil an unexpectedly diverse and likely well-adapted microbiome specific to the cryosphere, and supports the notion of the cryosphere as a biome with its distinct association of microorganisms, alongside plants and animals[17].

**Compositional patterns across cryospheric ecosystems.** We next explored how microbial community composition varies across cryospheric ecosystems. Using similarity analyses, we found that the microbiome composition differed significantly between cryospheric ecosystem types (PERMANOVA, $r^2 = 0.183$, $p < 0.001$; pairwise.adonis, $p < 0.001$ for all pairwise comparisons) (Fig. 3A and Supplementary Table 4). Most conspicuous was the segregation of snow/ice and marine communities, bracketing freshwater and terrestrial cryospheric communities. We also found a significant but relatively small effect of the primer pair (PERMANOVA, $r^2 = 0.019$, $p < 0.001$) that could be attributable to primer bias, or inherent differences related to sampling. To further assess these distributions, we explored prevalence patterns to identify a core microbiome across cryospheric ecosystems (Fig. 3B). We found 37 bacterial genera, including *Pseudomonas*, *Acinetobacter*, and *Flavobacterium*, for instance, to constitute the core microbiome. The disproportionate representation of these core genera in the above-identified cryospheric genera (Fisher's exact test, $p < 0.001$, odds ratio = 6.93) underlines their high abundance in cryospheric ecosystems (Supplementary Fig. 2). It also shows the prevalence and abundance of some cryospheric genera, indicating their potential relevance for ecosystem processes.

Additionally, analysing the relative abundance of the core cryospheric genera for each primer pair and cryospheric ecosystem types, we found that ice and snow microbiomes were associated with the highest proportions of core genera (23.05%

and 24.8% for PP1 and PP2, respectively) (Fig. 3D). In contrast, the marine cryospheric microbiome is only marginally composed of these genera (16.9% and 13.3% for PP1 and PP2, respectively). This pattern is in line with our unconstrained ordination analysis (Fig. 3B) and suggests that snow and ice represent endmember cryospheric systems, while the cryospheric component of the microbiome dissipates in downstream freshwaters, soils and the coastal ocean. Furthermore, the alpha-diversity was higher in terrestrial (Shannon $H = 3.67$), marine ($H = 3.25$) and freshwater ($H = 2.99$) ecosystems than in snow and ice ($H = 2.86$), corresponding to increasing contributions of ancillary taxa to their microbiomes (Supplementary Table 5). These differences in diversity are likely attributable to environmental gradients characterised by more diverse energy sources and niches, such as when moving from snow and ice to aquatic and soil ecosystems. Our analyses revealed compositional patterns of the cryospheric microbiome suggesting that snow and ice ecosystems including supraglacial habitats (e.g., mountain glaciers, ice sheets, snow and cryoconites) may serve as a source of cold-adapted bacterial diversity, upon losing which the downstream diversity may become threatened as well.

**Functional potential of the cryospheric microbiome.** The adaptive and survival strategies of microorganisms to the extreme environmental conditions of the cryosphere have received substantial attention over the last years[26–28]. For example, genomic insights from bacterial cultures have revealed mechanisms of thermal adaptation linked to bulk genomic features, such as GC content and genome size[29]. Moreover, genome streamlining has been shown to be a relevant evolutionary force in the cryosphere[28]. Therefore, we analysed the GC content and genome size of 13,414 reference genome sequences from the NCBI Refseq genomes database[30] to investigate shared properties of cryospheric genera, and to provide a framework to contrast future cryospheric metagenomic results. By comparing these reference genomes representing 660 bacterial genera present in our taxonomic analyses (29.8% of which are cryospheric genera according to our differential abundance analysis), we found that the cryospheric genera had a significantly higher GC content (Supplementary Fig. 3B; Wilcoxon test, Holm adj. $p = 0.0011$, median

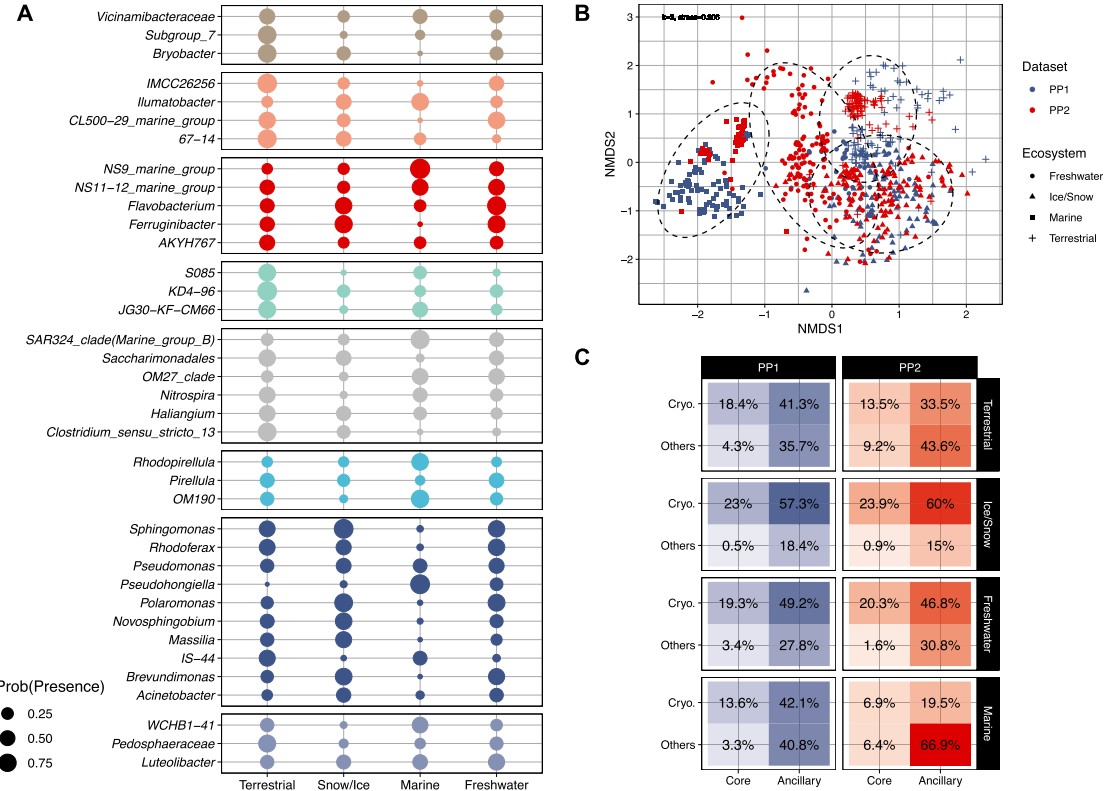

**Fig. 3 Microbiome structure across various cryospheric ecosystems. A** The probability of presence of members of the core microbiome is shown across cryospheric environments. Colours and facets separate phylum-level taxonomic affiliation. **B** Unconstrained ordination showing differences (Bray-Curtis dissimilarity) of bacterial communities among different cryospheric ecosystems ($k = 2$, stress $= 0.206$). Hulls demark 95% confidence intervals for a multivariate t distribution for the respective ecosystem types. **C** Mean relative abundance (in percentage) of core/ancillary and cryospheric/others bacterial genera across the four ecosystem types and the two primer pairs. Source data are provided as a Source Data file.

difference $= 8.8\%$) compared to the other genera; a pattern also supported by an enrichment in sequences that encode GC-rich amino acids (e.g., Alanine, Arginine, Glycine) (Supplementary Fig. 3A and Supplementary Table 6). Therefore, our findings suggest that cryospheric genera indeed share an elevated GC content[31], in line with reports on cold-adapted *Synechococcus* (SynAce01)[32] and *Actinobacteria*[33]. We also report that the average genome size of cryospheric genera is closely bracketed by published values for psychrophilic bacteria[34].

Next, using a gene-centric approach, we explored the functional space of the cryospheric metagenomes dataset. Out of 17,191 KEGG orthologues (KO), 980 KO were significantly enriched in cryospheric samples. Cryospheric genera and particularly cryospheric core members (e.g., *Pseudomonas*, *Sphingomonas* and *Novosphingobium*) disproportionately accounted for these gene families (Fig. 4A). Our analysis highlighted the relevance of chemolithotrophic pathways (e.g., manganese and iron uptake, sulfur, nitrogen and hydrogen metabolism), complementing earlier reports on these particular functional attributes of cryospheric ecosystems (Fig. 4B)[24,35,36]. The apparent relevance of chemolithotrophic pathways is likely attributable to a relative scarcity of organic carbon in cryospheric ecosystems. Interestingly, we consistently identified chitinase genes, which are involved in permafrost carbon cycling, but may also be an adaptation to freezing[37]. Finally, genes involved in adhesion, motility and various secretion systems collectively point to biofilm formation as an important strategy for life in cryospheric ecosystems[38], which are often characterised by extended periods of oligotrophy and elevated UV-radiation.

Our cross-ecosystem metagenomic analyses not only shed light on potential functions of the cryospheric microbiome across ecosystems, but also unveiled a large uncharacterised functional space with 43.4% of the protein coding genes in cryospheric samples unannotated to a KEGG orthologous group. While this does not seem unusual for environmental metagenomes in general[3], it is notable that we may lose this functional potential as the cryosphere vanishes. In order to shed light on this uncharacterised functional space, we clustered 41,068,842 gene sequences based on a 30% sequence similarity and 80% sequence coverage threshold, subsequently mapping representative sequences of the largest clusters (>29 sequences in at least 2 samples, $n = 12{,}125$) to the UniProt TrEMBL database (Fig. 4C). While the KEGG assigned clusters overall had a high percentage of sequences that matched genes in the UniProt database (Table 1), we found that cryosphere specific sequences show a large decrease in the clusters assigned to multiple KEGG (i.e., ambiguous) and even more in the ones containing exclusively unassigned sequences, compared to non-cryospheric environmental metagenomes. In addition to the low percentage of gene sequences matching UniProt sequences, we found that the cryosphere specific clusters that align to the database show a largely decreased identity with the matching sequence (Supplementary Fig. 4). These findings underline the lack of representation of cryospheric sequences in current gene sequences databases, potentially linked to the specificity of certain taxa to the cryosphere, and/or functions. Finally, the large nucleotide similarity within these clusters (Supplementary Fig. 4) suggests that these are conserved functions of particular importance to

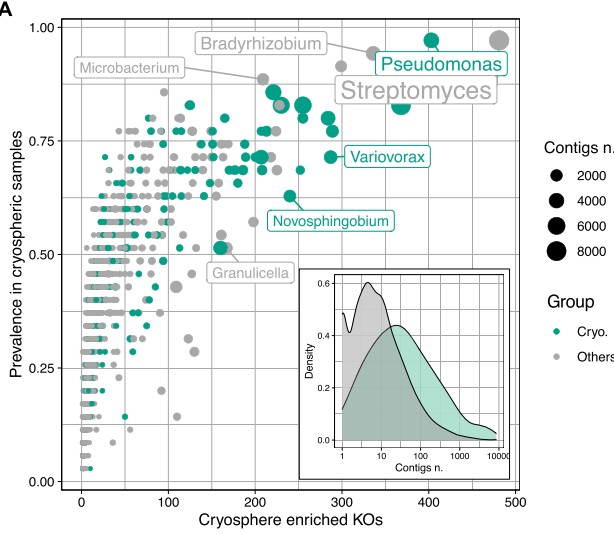

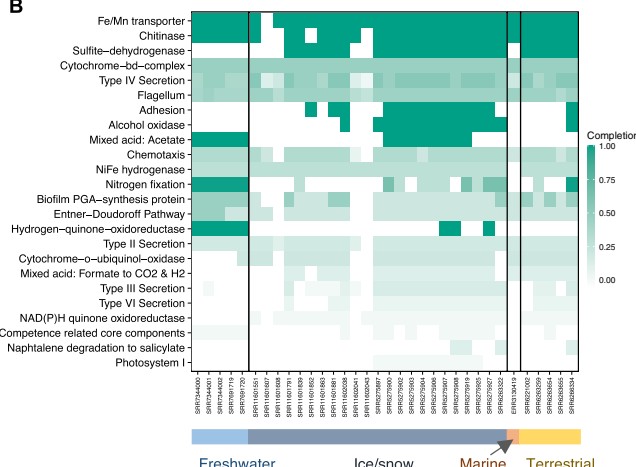

**Fig. 4 Functional enrichment analysis and taxonomy of enriched functions. A** Prevalence represented against the number of enriched KOs in cryospheric samples across bacterial genera. The shading represents the cryospheric and others bacterial genera whereas symbol size represents the number of contigs taxonomically assigned to the respective genus within cryospheric metagenomes. The insert represents the distribution of the number of contigs harbouring cryospheric enriched KOs across the cryospheric and others genera. **B** Heatmap representing the completion of pathways across cryospheric samples based only on the KOs enriched in the cryosphere. Source data are provided as a Source Data file.

microbial life under cryospheric constraints, and corroborates the notion of specific lineages of closely related taxa to dominate microbial life in the cryosphere. Aside from being uncharacterised, 170 of the unassigned gene clusters were only detected in cryospheric metagenomes and could thus represent unknown gene families of importance to understanding the adaptation of bacteria to these extreme ecosystems.

Collectively, our insights both at the taxonomic and functional level reveal key microbiome features that are exclusive to cryospheric ecosystems. Although entire taxonomic lineages are not unique to cryospheric ecosystems, it is evident that specific species and potentially strains are novel and adapted to these environments. Similarly, the emergent functional properties clearly demonstrate the exclusivity of functions, especially those that are yet to be characterised or that can be classified based on existing databases, within the cryosphere. On the contrary, we find that in both the taxonomic and functional complements,

**Table. 1 Description of the gene sequences clustering approach.**

| Annotation | Category | Number of clusters | Uniprot match (%) |
|---|---|---|---|
| KEGG | Cryosphere | 47 | 61.70 |
| | Shared | 1663 | 54.18 |
| | Non-cryosphere | 2325 | 55.14 |
| Ambiguous | Cryosphere | 113 | 40.71 |
| | Shared | 1056 | 52.65 |
| | Non-cryosphere | 3105 | 54.17 |
| Unassigned | Cryosphere | 170 | 17.65 |
| | Shared | 1524 | 5.18 |
| | Non-cryosphere | 2122 | 46.94 |

Table summarising the 12'125 largest gene sequence clusters present in at least two samples. The annotation refers to the assignment of the genes to one KEGG Orthologous group (KO), multiple KOs or unassigned (Ambiguous) and only unassigned (Unassigned). Distribution of assigned (KEGG), ambiguous and unassigned functional gene clusters highlighting the bias against cryospheric gene clusters. Shared refers to the representatives of both categories of samples contained gene sequences in the cluster. The number of clusters is shown, along with the proportion of clusters having a consensus sequence matching the UniProt database.

several taxa and functions are shared with non-cryospheric ecosystems. This is expected since the underlying genomic content supporting the taxonomic and functional annotations are shared between the cryospheric and non-cryospheric ecosystems. This is evident based on the >50% identity among the shared gene clusters that had matching identities in the KEGG database (Table 1).

Here we present what we believe is the first global data-driven approach to unravel specific features of the cryospheric microbiome. Our meta-analysis revealed diverse, distinct and functionally specific bacterial communities that appear to have been shaped by sustained evolutionary forces, suggesting an ancient origin of this biodiversity. While our study highlights key taxonomic groups such as *Proteobacteria* and *Bacteroidota*, our findings also disclose the importance of yet-uncultured bacteria and an uncharacterised genetic repertoire. In light of the threatened nature of the cryosphere, targeted efforts to unravel the phylogenetic and genomic underpinnings of bacterial adaptation to cryospheric ecosystems, including prospecting for cold-adapted biomolecules as well as the cultivation of cryospheric bacteria, are urgently required.

## Methods

**16S rRNA datasets.** Two primer pairs typically used in microbial ecology targeting the prokaryotic 16S rRNA were assessed: 341f-785r and 515f-806r. They will be referred to as Primer Pair 1 (PP1)[13] and Primer Pair 2 (PP2)[12], respectively. All articles citing the PP1 and PP2 reference articles were retrieved using Web of Science (All databases, searched on the 7 December 2019, 1727 articles). The first selection based on title and abstract was performed as described herein. Only studies having sequenced environmental samples were kept. Simultaneously, studies assessing pollution or contamination and involving major climatic or ecological events, e.g. storms or blooms, were removed. Thereafter, a second selection was performed based on the whole article, assessing technical criterions. Only studies using the aforementioned primer pairs, Illumina paired-end sequencing and having available data were kept; and the corresponding NCBI bioproject accessions were extracted. At a later stage, a few studies meeting the filtering criteria but not included in the Web of Science search were added.

The raw sequencing (fastq) data were downloaded using the ENA browser (European Nucleotide Archive; www.ebi.ac.uk/ena/browser/). At this stage, only the control samples were downloaded for experimental studies. The read files were filtered as follows: First, Trimmomatic was used to remove low quality reads, truncating the reads at the first instance of a sliding-window (5 bp) having a mean quality below 15[39]. At this stage, the raw data from each BioProject was imported into qiime2[40]. Denoising was performed with the dada2 plugin using the primers sequences length for the '-p-trim-left-r' and '-p-trim-left-f' parameters[41]. This step removed integrally two studies in the PP1 dataset ("negative values in quality" and "all samples discarded" errors). Only sequences assigned to bacterial taxa were

kept, and chloroplast and mitochondrial sequences were also removed. Finally, all samples with less than 5000 reads after this initial filtering were removed.

Taxonomy classification for PP1 and PP2 ASVs was performed using the qiime2 'feature-classifier' plugin and the Silva 138 nr99 database[41,42]. First, reads were extracted from the reference sequences using the extract-reads method. For this, the primer sequences were used for the '–p-r-primer' and '–p-f-primer' arguments. The length of the extracted reads was set to min. 250 and max. 450 for the PP1 dataset and min. 200 and max. 400 for the PP2 dataset. A classifier was then created using the fit-classifier-naïve-bayes method with the extracted reads and the reference taxonomy. Finally, this classifier was run on the dataset's sequences using the 'classify-sklearn' method to get the sequences taxonomy[41]. To keep only high-quality samples, all samples having <75% of their ASVs assigned to the phylum level, and 50% assigned to the genus level were removed. This filtering resulted in 2508 samples and 530,254 ASVs for PP1 and 1739 samples and 410,931ASVs for PP2. The ASV tables and metadata tables for these datasets can be found on Zenodo, under the file names: 'Data/PP1_table.tsv', 'Data/PP2_table.tsv' and 'Metadata/PP1_metadata.tsv' and 'Metadata/PP2_metadata.tsv', respectively.

**Metagenomic dataset.** To address the functional aspect of identified taxa, accession numbers of studies comprising of the following keywords: metagenomics, whole genome shotgun, and environmental, were queried using NCBI's EDirect[43]. The results were manually curated to select studies from a broad Geographic distribution, yielding a total of 382 datasets. The selection of metagenomic samples was further restricted to raw fastq data, thus precluding the use of samples from MG-RAST since only the metagenome assembly files were provided. Additionally, all samples still under embargo in accordance with the Joint Genome Institute (JGI; USA) policy, were excluded. From this collection, samples with fewer than 1 million reads or with a quality of reads less than Q25 were removed for a final collection of 91 samples (Fig. 1A). Paired reads were processed using the Integrated Meta-omic Pipeline (IMP)[44]. The workflow includes pre-processing such as primer/adaptor removal and trimming followed by an iterative assembly. Additionally, functional annotation of genes based on custom databases was performed (described below). The entire workflow is setup in a reproducible Snakemake format[45]. Briefly, after preprocessing the reads, de novo assembly using MEGAHIT (v1.2) assembler was performed[46]. All the methods and parameters used are listed on the Github repository, in the 'Preprocessing/IMP_config.yaml' file. The metagenomic dataset KEGG Orthologs (KO) table, taxonomy table, and metadata are available on Zenodo under the 'Data/MTG_KEGG_counts.tsv', 'Data/MTG_table.tsv', and 'Metadata/MTG_metadata.tsv'.

**Metagenomic taxonomic classification and functional analyses.** Functional potential analyses from contigs were determined by predicting open-reading frames using a modified version of Prokka[47] including Prodigal[48] gene predictions for complete and incomplete open reading frames. Genes identified subsequently were annotated with Hidden Markov Models (HMM)[49], trained using an in-house database[50]. The annotations were further annotated with KO[51] groups using 'hmmsearch' from HMMER 3.1[49]. Upon multiple functional group assignments, the best hits based on bit scores were selected. FeatureCounts[52] with the '-p' and '-O' arguments were then used to extract the number of reads per functional category.

**Logistic regression classification of cryospheric bacterial communities.** The Logistic regression implemented in scikit-learn python module (version 0.23.2) was trained on presence-absence ASV tables to classify cryospheric samples[53]. To reduce the amount of ASVs considered, the table was filtered based on relative abundance: presence was defined at a 0.005 relative abundance threshold. A stratified 5-fold cross-validation (CV) was ran and the scores were averaged across the CVs. This process was repeated 40 times and the mean and standard deviations are reported for each metric. To ensure reproducibility, the seed was set as 23 for the classifier, and as the iteration number for the stratified cross validation iterator (from 0 to 39). The C parameter controlling L2 penalisation was turned off using the 'none' argument and the lbfgs solver was used. ROC curves were plotted using the 'plot_roc_curve' function of the scikit-learn python module. Balanced accuracy, precision and recall were computed using the 'accuracy_score', 'precision_score' and 'recall_score' methods, respectively, with sample weights correcting for the sample size of the cryospheric and non-cryospheric datasets (Supplementary Table 1). The means and standard deviations of scoring metrics for the classifiers can be found in table S1. Odds ratios were calculated using the exponent of the logistics models coefficients. The tables containing the ASVs logistic regressions odds ratios can be found in the Data folder available on Zenodo under the name 'PP1_Logistic_coefs.csv' and 'PP2_Logistic_coefs.csv' for PP1 and PP2, respectively.

**Phylogenetic analyses.** Phylogenetic trees were built using the set of ASVs found in the dataset used for the logistic regression classification. Due to the different 16 S regions targeted, phylogenies for both PP1 and PP2 datasets were constructed separately. The ASVs sequences were aligned using the FFT-NS-2 algorithm implemented in the Mafft aligner with default parameters[54]. The alignments were subsequently trimmed using TrimAl with the '-gt 0.95' parameter, and the trees built using IQ-TREE with the GTR model of nucleotide substitution and the '-fast' option[55,56]. Phylogenetic tree visualisations were created using the ggtree and

ggtreeExtra R packages[57,58]. Only positive coefficients showing enriched presence in cryospheric environments are shown in the phylogenetic barplots (Fig. 1). The number of ASVs with an odds ratio above 1 was shown for taxonomic summaries (Supplementary Fig. 1B, C).

ß-diversity phylogenetic metrics (Sorensen's Index and ß-MNTD) were computed using the 'phylosor' and 'comdistnt' functions of the Picante R package[59], using custom functions to compute pairwise comparisons. For each metric, 50 iterations were performed where we calculated the pairwise distances between and within 50 cryospheric, and 50 non-cryospheric samples. This random sub-sampling approach was chosen to reduce computing time. Kruskal–Wallis tests were used to determine whether the distribution was different across groups, and Wilcoxon tests were used to calculate pairwise post-hoc comparisons. Wilcoxon tests implemented in the compare_means function of the ggpubr R package were used, effects sizes (r) were calculated with the wilcox_effsize function implemented in the statix R package. Sample specific calculations of α-PD (and species richness), α-MPD and α-MNTD were computed using the 'pd', 'mpd' and 'mntd' functions of the Picante R package[59]. Linear models were used to compare the values of α-PD, α-MPD, and α-MNTD across samples, taking the logarithm of the species richness and the dataset (PP1 and PP2) as a fixed effect.

**Differential abundance analysis.** Using the Silva Taxonomic information[42], ASV raw counts were aggregated to the genus-level using a custom R script, removing the ASVs not assigned taxonomically to the genus-level. Ancom v2.1 was used on the count data for the differential abundance analysis, using the default W statistic threshold of 0.7[60]. The 'zero-cut' parameter was set to 0.995 to consider all bacterial genera present in at least 21 samples ($n = 4247$), and the primer pair (PP1 and PP2) variable was taken as a random effect with the rand_formula parameter ("~1| Dataset"). We considered significantly enriched genera (i.e. cryospheric genera), the ones with a W statistic above the threshold (0.7) and a positive value of CLR mean difference. GGplot2 was used to modify the Ancom v2.1 figure showing the results of the differential abundance analysis. The 'heat_tree' function of the metacoder R package (version 0.3.4) was used to show the number of cryospheric bacterial genera, at various taxonomic level, using taxonomic trees[61]. The results of this analysis can be found in the Data/ folder available on Zenodo under the name 'Ancom_amplicon_res.csv' file.

**NCBI Refseq genomes properties.** To assess the genome size and GC content of publicly available prokaryote genomes, a non-redundant list encompassing all the genera in our datasets was compiled. Thereafter, the list of prokaryote genomes (prokaryotes.txt) available on NCBI[30] was downloaded on March 15th, 2021 from https://ftp.ncbi.nlm.nih.gov/genomes/GENOME_REPORTS. The prokaryote list was filtered based on the list of genera found in our dataset, simultaneously retrieving the accession IDs. The accession IDs were used to download the complete bacterial genome sequences using the ncbi-genome-download python package (https://github.com/kblin/ncbi-genome-download). The genome sizes for the downloaded genomes were additionally retrieved from the prokaryotes.txt metadata file. Subsequently, Prodigal[48] was used to annotate the open-reading frames per genome obtaining both the general feature format (gff) files and aminoacid fasta (faa). These were used thereafter as input used to estimate the predicted growth time (in hours) and their codon usage analyses (CUB) using gRodon and coRdon (https://github.com/BioinfoHR/coRdon) R package respectively[62,63]. The amino acid enrichment analysis was performed on by converting the codon counts to amino acids using the R-package Biostrings using DEseq2 with default parameters (log-median ratio normalisation across genera). Wilcoxon tests implemented in the compare_means function of the ggpubr R package were used, effects sizes (r) were calculated with the wilcox_effsize function implemented in the statix R package. The relevant scripts and information for these analyses are openly available and included in the code availability section. The corresponding files used for this analysis can be found in the Data/ folder available on Zenodo under the names 'prokaryotes.txt', 'merged_all_codon_counts.txt' and 'merged_all_growth_prediction.txt'.

**Structure of the cryospheric microbiome.** Non-metric multidimensional scaling was used to visualise the composition of cryospheric bacterial communities according to the ecosystem types and primer pairs. For this, the 'metaMDS' function implemented in the package vegan was used with Bray-Curtis distances[64]. The stress for the chosen value of $k = 2$ was 0.206. The 'adonis2' function was used to perform a PERMANOVA analysis to test the effect of the ecosystem type and the primer pairs on the composition of bacterial communities (Supplementary Table 4). Pairwise comparisons between ecosystem types were tested using the function 'pairwise.adonis2'[65]. P-values were adjusted using the default Bonferroni method, to account for multiple comparisons.

The prevalence of each genus was modelled as the probability of presence using a logistic binomial regression (with the R stats 'glm' method), using the ecosystem type (snow/Ice, terrestrial, marine and freshwater) and the primer pair as fixed effect. To calculate the probability of occurrence in the cryosphere for each genus, the prediction was calculated for all ecosystem types and primer pair combinations, and averaged. The core microbiome was defined at 0.1% abundance, and 20% prevalence across the cryosphere, for genera present in at least one sample in all

four ecosystem types (Supplementary Fig. 2B). The core microbiome presence in the different ecosystem types was shown using an upset-plot using the *complex-upset* R package[66]. The taxonomic tree available in Supplementary Fig. 2A was created using the *Metacoder* R package[61]. The α-diversity was calculated using Shannon's index with a custom R functions[67]. To test the difference across ecosystems and datasets, the Wald-Type statistics implemented in the 'GFD' function of the R *GFD* package was used (Supplementary Table 5). This test was performed instead of an ANOVA, as the data was not normally distributed. The mean values given by the function were used for the ecosystem comparison.

**KEGG enrichment**. The standard DESeq2 pipeline with default parameters was used on raw KEGG counts for the enrichment analysis, using the default Wald tests[51,68]. We considered enriched Kegg Orthologs (KOs) with an FDR adjusted $p < 0.01$ and a $\log_2$ fold-change >1. To unravel the contribution of these gene families to functional pathways, we ran *KEGGdecoder*[69] on the KOs enriched in cryospheric samples, to identify environmental-associated pathways in all samples.

To understand and unravel the origins of the gene families specific to the cryospheric metagenomes, contigs were taxonomically classified following which the specific gene families were mapped to the contigs. We used *Kraken2* to taxonomically assign all the contigs present in the metagenomes followed by custom python scripts (provided) to link the genes belonging to enriched KEGG orthologs (KO and the corresponding NCBI taxon ID[70,71]. An R script using the NCBI *entrez* package was used to retrieve the taxonomy based on the taxon ID, and to get the genus-level taxonomy[43]. To link the Silva genus taxonomies with their NCBI counterparts, the grep function included in R allowing partial matches was used to find Silva genera name matching the NCBI genus name. The DESeq2 results, KEGG-decoder output and taxonomy matches can be found in the Data/ folder of the Zenodo under the names 'KEGG_deseq_results.csv', 'KEGG_decoder_output', and 'KEGG_sign_tax_genera.csv', respectively.

**Gene clusters and unassigned protein coding sequences**. Predicted gene sequences annotated to the KEGG database and those unassigned were gathered into individual groups based on KEGG ID or Unassigned using a custom python script. 'annotation2gene.py'. The fasta files were subsequently concatenated and clustered to identify functional homologues within the dataset. We used *mmseq2* 'linclust'[72] to cluster the 41,068,842 gene sequences found in the entire metagenomic dataset. Subsequently, fasta sequences associated with each cluster were retrieved into separated clusters ($n = 12,125$) and linked to the coverages to estimate abundances. *MAFFT*[54] was then used to create a multiple sequence alignment of the sequences per cluster, while the 'cons' method from *EMBOSS* was used to generate a consensus sequence. The generated consensus sequences from each cluster were subsequently annotated and their identity verified against the UniProt TrEMBL[73] database. The pairwise identity of sequences within each cluster was measured using *CLUSTAL*[74] 'distmat' option with the '–percent-id'. Wilcoxon tests implemented in the *compare_means* function of the *ggpubr* R package were used, effects sizes ($r$) were calculated with the *wilcox_effsize* function implemented in the *statix* R package. The unassigned clusters summary statistics and Uniprot matches can be retrieved on Zenodo, in the Data/ folder under the names 'Unassigned_-clusters_stats.tsv', and 'unassigned_uniprot_matches.txt'.

**Reporting summary**. Further information on research design is available in the Nature Research Reporting Summary linked to this article.

## Data availability

The data generated in this study have been deposited in Zenodo, under https://doi.org/10.5281/zenodo.6541278. Source data used for figures are provided with this paper.

## Code availability

All scripts used for analyses, along with the conda environments, and additional information is provided in a Github repository archived on Zenodo: https://doi.org/10.5281/zenodo.6587400.

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

## Acknowledgements

This work was supported by The NOMIS Foundation (Vanishing Glaciers) to T.J.B.; S.B.B. was supported by a Swiss National Science Foundation (CRSII5_180241) grant to T.J.B and P.W. We extend our gratitude to Laura de Nies, Patrick May, Cedric Laczny, and Valentina Galata for their advice on metagenomic analyses. Computational work was carried out using the HPC facilities of the University of Luxembourg.

## Author contributions

M.B.: conceptualisation, methodology, investigation, formal analysis, data curation and writing – original draft preparation, visualisation; S.B.B.: conceptualisation, methodology, investigation, data curation and writing – original draft preparation; S.F.: conceptualisation, methodology, data curation and writing; H.P.: conceptualisation, methodology, data curation and writing; A.W.: conceptualisation and methodology; T.K.: data curation and writing; L.E.: methodology; G.M.: methodology; P.W.: conceptualisation and supervision; T.J.B.: conceptualisation, writing – original draft preparation, data curation, supervision and funding acquisition.

## Competing interests

The authors declare no competing interests.
