## [Peer Review File · Nature Communications]

Reviewers' Comments:

Reviewer #1:

Remarks to the Author:

This manuscript is a well written and well thought out piece of work that reanalyzed existing cryosphere taxonomic (16SrRNA gene) and functional (metagenomic) data and compared it with data from non-cryosphere ecosystems to identify characteristics unique to cold dwelling bacteria. It is important to catalogue these organisms and functions and the bioinformatic and statistical approaches are appropriate and lead to results that support existing published data. The main findings include the identification of a taxonomically diverse group of bacteria that seem to be consistently found across cryosphere environments and the existence of undescribed functions within the metagenomes. While this work has merit and is of importance to the research community, it is unclear what the main novelty is. For example, one of the more interesting ideas put forward in this manuscript is related to the evolutionary history of cryosphere microorganisms, that they are potentially ancient, but it is unsupported by the data presented.

Specific concerns outlined in Line comments.

1. Lines 53-55: The authors suggest that the catalogue of organisms and functions they have curated can be used as a foundational resource for the study of cryospheric life. This has the potential to be useful, but one issue that might limit this is the lack of details on the metadata and sampling sites and seasons. Although presented in a figure (figure 1), it would have been useful to summarize this in a table form so that the reader can have a better idea of spatial and ecosystem representativity without going through the list of source articles provided. From the figure, it seems that there is a strong bias towards studies in the Arctic and specifically Svalbard, and from a few sites in Antarctica. There do not appear to be many studies on alpine ecosystems either, is this because the authors consider that mountain ecosystems fall outside the definition of the cryosphere or that there is a lack of relevant data to include them?
2. Lines 67-70: I understand the difficulty in obtaining high quality metagenomes, but can 34 metagenomes provide a global picture of the functional characteristics of the cryosphere communities that exist in multiple niches all over the globe? It might be useful to address this in the discussion.
3. Lines 74-75: I agree that this a relatively comprehensive data set, however, it seems like a missed opportunity to perhaps comment on cryosphere ecosystems and niches that weren't included due to the lack of data.
4. Lines 100-110: This is an interesting idea, but some of the statements are difficult to support with available data, specifically with regard to contemporary evolution and assembly processes. A number of articles have looked at transduction, horizontal gene transfer and selection processes in cryosphere ecosystems (e.g. Vollmers et al., 2013, Sanguino et al., 2015, Ciok et al., 2018, Dorrell et al., 2021, Rapp et al., 2021, Zhong et al., 2021) and suggest that they are critical for colonization and survival in the cryosphere. Please consider rephrasing this.
5. Lines 118-120: This is a little bit confusing, in the sentences above, the authors state that cryosphere bacteria are found within a high diversity of phyla, but then base their assumption for early speciation events on the observation that the cryosphere genera mostly fall within 2 phyla, Proteobacteria and Bacteroidota. These phyla are highly diverse and dominant within these ecosystems, so it is not surprising that many cryosphere genera are found within them. Also, this does not preclude assembly and genomic rearrangement or HGT events.
6. Lines 165-177: This result is interesting and highlights the urgency for studying cryosphere ecosystems. I was wondering to what extent the core cryosphere microbiome of snow and ice contributes to downstream ecosystem diversity. The quality of figure 3 should be improved for better readability.
7. Lines 190-192 and extended figure 3: Please be careful of the language used. The genomes contain a higher abundance of sequences that encode GC rich amino acids, since amino acids were not measured. Also in figure 3B, please consider rephrasing B) The median codon usage bias (ConsistencyHE) of the highly-expressed genes (eg. ribosomal, 22 gyrase and other housekeeping genes), since gene expression was not measured.

Reviewer #2:

Remarks to the Author:

Bourquin et al. report on the community structure, phylogenetic, taxonomic and functional analysis of microbes in the cryospheric ecosystems. The authors obtained 510 published 16S rRNA gene samples from cryospheric ecosystems, including polar ice sheets, mountain glaciers, proglacial lakes, permafrost soils, and the coastal ocean under the influence of glacier runoff, and compared with published 16SrRNA gene samples from non- cryospheric ecosystems. In addition, 34 published metagenomes from cryospheric ecosystems with 56 metagenomes from similar but non-cryospheric ecosystems were collected and compared, including 2,427,818,072 paired reads yielded 41,068,842 gene sequences. This manuscript is an interesting topic and may contribute novel information into the field of cryospheric ecosystems. However, the short length of the two 16SrRNA primer pairs, limited sample size and metagenomic raw data hindered the significance of this manuscript.

Major comments:

1. The products of both 16S rRNA primer pairs are short and not suitable for phylogenetic and taxonomic analysis.
2. The sample number of 16S rRNA genes from the cryospheric ecosystems is limited and collected about two years ago (December 2019). The new dataset should be added, especially for the 16S rRNA genes, which is easy to analysis.
3. The sample size of metagenomic raw data from cryospheric ecosystems is limited. Hence, the results might not be able to represent the whole cryospheric ecosystems. Some new dataset should be added.
4. Actually, most of the bacterial genera and functional gene clusters in the cryospheric ecosystems are shared with other non- cryospheric ecosystems. Whether the author could state that the microbiome in the cryospheric ecosystems is novel? The linkage between cryospheric ecosystems and non- cryospheric ecosystems should be addressed.
5. Fig. 3B should be revised.

Manuscript Reference: NCOMMS-21-44625-T

Response to the Editor and reviewer's comments.

Reviewer #1:

This manuscript is a well written and well thought out piece of work that reanalyzed existing cryosphere taxonomic (16SrRNA gene) and functional (metagenomic) data and compared it with data from non-cryosphere ecosystems to identify characteristics unique to cold dwelling bacteria. It is important to catalogue these organisms and functions and the bioinformatic and statistical approaches are appropriate and lead to results that support existing published data. The main findings include the identification of a taxonomically diverse group of bacteria that seem to be consistently found across cryosphere environments and the existence of undescribed functions within the metagenomes. While this work has merit and is of importance to the research community, it is unclear what the main novelty is. For example, one of the more interesting ideas put forward in this manuscript is related to the evolutionary history of cryosphere microorganisms, that they are potentially ancient, but it is unsupported by the data presented.

Answer: We would like to thank this reviewer for her/his overall positive evaluation of our work. We believe that our work is novel as it is the first to present an encompassing analysis across various cryospheric ecosystems, including a robust comparison with similar but non-cryospheric ecosystems. To the best of our knowledge, this has not been published before. It is also our multi-faceted approach, including taxonomic, phylogenetic and metagenomic analyses, that sheds new light on the cryospheric microbiome. Please, see our detailed responses below.

Specific concerns outlined in Line comments.

1.1. Lines 53-55: The authors suggest that the catalogue of organisms and functions they have curated can be used as a foundational resource for the study of cryospheric life. This has the potential to be useful, but one issue that might limit this is the lack of details on the metadata and sampling sites and seasons. Although presented in a figure (figure 1), it would have been useful to summarize this in a table form so that the reader can have a better idea of spatial and ecosystem representativity without going through the list of source articles provided. From the figure, it seems that there is a strong bias towards studies in the Arctic and specifically Svalbard, and from a few sites in Antarctica. There do not appear to be many studies on alpine ecosystems either, is this because the authors consider that mountain ecosystems fall outside the definition of the cryosphere or that there is a lack of relevant data to include them?

Answer: We are grateful for your suggestion to add more metadata. We have done so now by adding an extensive table to the Extended data Table 7.

We agree that our samples were biased towards the polar regions. Obviously, this is a result of the available data from literature, and fitting our strict (and therefore conservative) criteria for retaining data for our analyses. It is clear that high-altitude ecosystems are part of the cryosphere. However, at the time when we collected data, only relatively few high-quality studies were available.

Given the uneven geographic distribution (high-latitude versus high-altitude), more studies from high-altitude ecosystems would be useful. As asked by reviewer #2, we have now added nine recent amplicon studies including 185 new cryospheric samples (4 additional alpine studies). We have also expanded in the text (lines 56 and 59) on the availability of data from diverse ecosystems.

1.2. Lines 67-70: I understand the difficulty in obtaining high quality metagenomes, but can 34 metagenomes provide a global picture of the functional characteristics of the cryosphere communities that exist in multiple niches all over the globe? It might be useful to address this in the discussion.

Answer: We thank the reviewer for this comment and acknowledge the concerns. In view of the comment, we have toned down the utility of 34 metagenomes in providing a ‘global’ overview, and have subsequently revised the manuscript in lines 72-74: “On the other hand, several niches such as glacier snow, glacier-fed rivers/streams, and the full-breadth of permafrost may not entirely be represented due to data unavailability.”. We have further expanded the discussion to reflect on the snapshot our functional characterisation provides with respect to the global cryosphere, as suggested.

1.3. Lines 74-75: I agree that this a relatively comprehensive data set, however, it seems like a missed opportunity to perhaps comment on cryosphere ecosystems and niches that weren’t included due to the lack of data.

Answer: We are grateful for this comment and as we responded in comment 1.1, we have expanded the manuscript in lines 48-49 and 60-65 to discuss cryospheric niches that were not included. The Extended data table summarising the cryospheric samples (1.1) also gives an overview of the niches that were included in our dataset.

1.4. Lines 100-110: This is an interesting idea, but some of the statements are difficult to support with available data, specifically with regard to contemporary evolution and assembly processes. A number of articles have looked at transduction, horizontal gene transfer and selection processes in cryosphere ecosystems (e.g. Vollmers et al., 2013, Sanguino et al., 2015, Ciok et al., 2018, Dorrell et al., 2021, Rapp et al., 2021, Zhong et al., 2021) and suggest that they are critical for colonization and survival in the cryosphere. Please consider rephrasing this.

Answer: This is a good point and we have rephrased the statements accordingly. Additionally, we also address the role of transduction, horizontal gene transfer events and selections processes as highlighted by the reviewer with the appropriately suggested references. The revised text can be found in lines 109-111.

1.5. Lines 118-120: This is a little bit confusing, in the sentences above, the authors state that cryosphere bacteria are found within a high diversity of phyla, but then base their assumption for early speciation events on the observation that the cryosphere genera mostly fall within 2 phyla, Proteobacteria and Bacteroidota. These phyla are highly diverse and dominant within these ecosystems, so it is not surprising that many cryosphere genera are found within them. Also, this does not preclude assembly and genomic rearrangement or HGT events.

Answer: Thanks for your critical eye. We agree that this was confusing and we apologise for it. The differential abundance analysis has identified 589 genera as over-represented in the cryospheric samples; indeed, they distribute over 49 phyla that span the bacterial tree. However, a large number of these genera were affiliated with *Proteobacteria* and *Bacteroidota*. To disentangle what appeared confusing, we have removed the sentence (In this context, our findings posit the hypothesis that...) and combined the paragraph here discussed with the next paragraph.

1.6. Lines 165-177: This result is interesting and highlights the urgency for studying cryosphere ecosystems. I was wondering to what extent the core cryosphere microbiome of snow and ice contributes to downstream ecosystem diversity. The quality of figure 3 should be improved for better readability.

Answer: We thank the reviewer for this comment. Molecular analyses have indeed revealed that glacier-fed rivers and the biodiversity therein are influenced by turnover between active and dominant taxa, which are potentially recruited from upstream sources ¹. Although in this case, the soils, rocks and groundwater, were the major driving factors, it is plausible that the core taxa from snow and ice microbiomes may indeed be contributing factors to the overall diversity of downstream ecosystems. Interestingly, the core microbiome of Snow/Ice defined as the bacterial genera present in at least a 1/5 sample at an abundance threshold of 0.001 in the Snow/Ice samples (78 genera) represents on average 2.7% of the marine communities, 25.9% of the freshwater, and 24.5% of the terrestrial cryospheric samples, while accounting for 68.6% of the Snow/Ice bacterial communities. An upset-plot of the overlap of the core microbiome of the four cryospheric ecosystem types has also been added to the extended data. As suggested an updated version of Figure 3 has been included in the revised manuscript. We apologise for the quality of Figure 3 in the initial submission.

1.7. Lines 190-192 and extended figure 3: Please be careful of the language used. The genomes contain a higher abundance of sequences that encode GC rich amino acids, since amino acids were not measured. Also in figure 3B, please consider rephrasing B) The median codon usage bias (ConsistencyHE) of the highly-expressed genes (eg. ribosomal, 22 gyrase and other housekeeping genes), since gene expression was not measured.

Answer: We acknowledge the reviewer's comment and have adjusted the language as suggested: "The median codon usage bias (ConsistencyHE) of housekeeping genes (eg. ribosomal, 22 gyrase etc.)".

Reviewer #2:

Bourquin et al. report on the community structure, phylogenetic, taxonomic and functional analysis of microbes in the cryospheric ecosystems. The authors obtained 510 published 16S rRNA gene samples from cryospheric ecosystems, including polar ice sheets, mountain glaciers, proglacial lakes, permafrost soils, and the coastal ocean under the influence of glacier runoff, and compared with published 16SrRNA gene samples from non- cryospheric ecosystems. In addition, 34 published metagenomes from cryospheric ecosystems with 56 metagenomes from similar but non-cryospheric ecosystems were collected and compared, including 2,427,818,072

paired reads yielded 41,068,842 gene sequences. This manuscript is an interesting topic and may contribute novel information into the field of cryospheric ecosystems. However, the short length of the two 16SrRNA primer pairs, limited sample size and metagenomic raw data hindered the significance of this manuscript.

Answer: We are pleased to read that the reviewer finds our manuscript interesting and potentially contributing to the field. Please see our detailed comments below.

Major comments:

2.1. The products of both 16S rRNA primer pairs are short and not suitable for phylogenetic and taxonomic analysis.

Answer: We agree with the reviewer on this point. Being fully aware of these limitations regarding the sequenced regions but also the 16S rRNA gene as a whole, we took extra caution to limit taxonomic resolution down to the genus level² and to focus our phylogenetic analyses to community-level phylogenetic turnover as it is very often the case for such datasets^{3,4}. This was indeed one of the reasons why we included two different primer pair sets in our analyses; the fact that we find similar taxonomic and phylogenetic patterns with both primer sets greatly substantiates our findings. In this context, we want to highlight that not many other meta-analyses have done the effort to compare patterns between various primer pairs. The use of genus-level taxonomy for all taxonomic analyses is discussed on lines 65-66.

2.2. The sample number of 16S rRNA genes from the cryospheric ecosystems is limited and collected about two years ago (December 2019). The new dataset should be added, especially for the 16S rRNA genes, which is easy to analysis.

Answer: We are grateful for this comment. Indeed, we are living in a high-speed era with new studies being published at rapid pace. Our initial submission reflected the state when we passed existing data through our very selective filter. Now, we have conducted a new literature search on papers published since our initial data collection and based on our previous criteria. As a result, we were able to add an additional nine amplicon studies including 185 new cryospheric samples, including for instance glacier-fed streams from New Zealand, permafrost from an alpine floodplain in Italy, and a glacier in Spain. To enlarge the data set even more, we have also included unpublished sequences from glacier-fed streams in the Caucasus that are from our lab.

2.3. The sample size of metagenomic raw data from cryospheric ecosystems is limited. Hence, the results might not be able to represent the whole cryospheric ecosystems. Some new dataset should be added.

Answer: We acknowledge the reviewer's concerns regarding the limited sample size for the metagenomic raw data. As mentioned in our response to reviewer #1 (point 1.2; now revised in lines 73-74), we have toned down the sense of using 34 metagenomic samples to give a global catalogue; we understand that this is rather a snapshot. However, at the time of submission, and also as highlighted in the Methods section of the manuscript, we retrieved all publicly available metagenomic datasets that matched our quality criteria. Of the identified studies, several samples were not included for the following reasons:

1. Raw FASTQ files had shallow sequencing efforts (<1 Mio reads).
2. Raw data was still under embargo on hosted repositories such as JGI.
3. Metagenomic data available on MG-RAST is only accessible in the assembly format and does not provide FASTQ files.

Moreover, the data used in our studies is a snapshot of global cryospheric ecosystems as indicated in the response to comment #1.2, and represents the status quo of publicly available data. Given the already comprehensive analyses in the original manuscript, we believe that reanalysing the now available data would most likely not add much to our current findings. Furthermore, since these analyses extremely time consuming, the availability of new metagenomes in the new future may raise again doubts on the actuality of our data set.

2.4. Actually, most of the bacterial genera and functional gene clusters in the cryospheric ecosystems are shared with other non- cryospheric ecosystems. Whether the author could state that the microbiome in the cryospheric ecosystems is novel? The linkage between cryospheric ecosystems and non- cryospheric ecosystems should be addressed.

Answer:

We thank the reviewer for this insightful comment. As highlighted in our findings, we find some specific features that are indeed unique to the cryosphere compared to the non-cryospheric ecosystems. For example, just a small fraction of the ASVs found in the cryosphere are shared with non-cryospheric ASVs. This is highlighted by the logistic regressions models that differentiate with great accuracy (> 0.97) cryospheric from the non-cryospheric microbiome. We also find unique gene clusters in the functional analysis, despite the low identity threshold for the clustering. However, for the taxonomic analyses, we have chosen to work with genus level taxonomy (please, see 2.1). At this level, the bacterial genera that we highlight are indeed not unique to the cryosphere. Good examples are *Flavobacterium*, *Polaromonas*, etc., which are cosmopolitan genera found in many locations and habitats across the globe. We have furthermore expanded the discussion to elaborate on the linkage between the cryospheric and non-cryospheric ecosystems with respect to shared taxa and functional gene clusters. The revised section is in lines 248-253.

2.5. Fig. 3B should be revised.

Answer: As indicated in the response to comment #1.7, we have revised Figure 3.

References

1. Wilhelm, L., Singer, G. A., Fasching, C., Battin, T. J. & Besemer, K. Microbial biodiversity in glacier-fed streams. *ISME J.* **7**, 1651–1660 (2013).
2. Singer, E. *et al.* High-resolution phylogenetic microbial community profiling. *ISME J.* **10**, 2020–2032 (2016).
3. Stegen, J. C. *et al.* Quantifying community assembly processes and identifying features that impose them. *ISME J.* **7**, 2069–2079 (2013).

4. Ning, D., Deng, Y., Tiedje, J. M. & Zhou, J. A general framework for quantitatively assessing ecological stochasticity. *Proc. Natl. Acad. Sci.* **116**, 16892–16898 (2019).

Reviewers' Comments:

Reviewer #1:

Remarks to the Author:

My initial opinion about this manuscript has not changed; it is a well written and well thought out piece of work that reanalyzed existing cryosphere taxonomic (16SrRNA gene) and functional (metagenomic) data and compared it with data from non-cryosphere ecosystems to identify characteristics unique to cold dwelling bacteria.

I have now gone through the updated manuscript and would like to thank the authors for integrating the comments raised. The additions and modifications have clarified some of the sticking points, and I appreciate the toning down of the language.

I thought that extended table 7 was particularly beneficial, and this new information raises some new questions. I was wondering about the selection criteria used for attributing the ecosystem type to the samples. Specifically, given the breadth of research on cryoconite holes and the uniqueness of their communities, both in terms of structure and abundance, do they really qualify as ice/snow ecosystems? Could this have biased some of the observations?

This may be out of the scope of the paper, but out of curiosity, do the cryosphere communities group together based on their ecosystem type?

These are minor questions, more out of interest.

Manuscript Reference: NCOMMS-21-44625B

Response to the Editor and reviewer's comments.

Reviewer #1:

My initial opinion about this manuscript has not changed; it is a well written and well thought out piece of work that reanalyzed existing cryosphere taxonomic (16SrRNA gene) and functional (metagenomic) data and compared it with data from non-cryosphere ecosystems to identify characteristics unique to cold dwelling bacteria.

I have now gone through the updated manuscript and would like to thank the authors for integrating the comments raised. The additions and modifications have clarified some of the sticking points, and I appreciate the toning down of the language.

I thought that extended table 7 was particularly beneficial, and this new information raises some new questions. I was wondering about the selection criteria used for attributing the ecosystem type to the samples. Specifically, given the breadth of research on cryoconite holes and the uniqueness of their communities, both in terms of structure and abundance, do they really qualify as ice/snow ecosystems? Could this have biased some of the observations?

This may be out of the scope of the paper, but out of curiosity, do the cryosphere communities group together based on their ecosystem type?

These are minor questions, more out of interest.

Response: We thank the reviewer for acknowledging the modifications and our efforts. We have defined the cryosphere *sensu lato*, hence different habitats (water, sediments, etc.) are grouped together (in the freshwater, marine, ice/snow and terrestrial ecosystem types). Indeed, cryoconites represent a unique habitat/niche, but based on the NMDS analysis (Figure 3B), the cryoconites cluster with the other Ice/Snow samples that include glacier ice, snow, cryoconites, etc. Furthermore, the NMDS validates our initial classification of ecosystem types which is additionally supported by statistical significance as highlighted in lines 145-148 in the revised manuscript. We have now included this description in lines 174-175 in the revised manuscript to further clarify for the reader.